# A Rare-Earth Free Magnetically Geared Generator for Direct-Drive Wind Turbines

**Reza Zeinali and Ozan Keysan** * 

Electrical and Electronics Engineering, Middle East Technical University (METU), Ankara 06800, Turkey; r.zeinali@tue.nl
**\*** Correspondence: keysan@metu.edu.tr; Tel.: +90-312-210-7586

**Abstract:** A novel Vernier type magnetically geared direct-drive generator for large wind turbines is introduced in this paper. Conventional Vernier-type machines and most of the direct-drive wind turbine generators use excessive amount of permanent magnet, which increases the overall cost and makes the manufacturing process challenging. In this paper, an electrically excited (PM_less) claw-pole type Vernier machine is presented. This new topology has the potential of reducing mass and cost of the generator, and can make the construction easy in manufacturing and handling. Analytical designs are verified using 3D finite-element simulations and several designs are evaluated to find the optimum design for a 7.5 MW, 12 rpm wind turbine application. It is shown, that the required torque can be achieved with an outer diameter of 7.5 m, and with a mass of 172 t (including the structural mass). The proposed generator is compared with commercial direct-drive generators, and it is found that the proposed generator has the highest torque density with 34.7 kNm/t.

**Keywords:** direct-drive generators; magnetic-geared generators; Vernier type machines; wind turbines

## 1. Introduction

Wind energy has the potential of playing an important role in the future electricity generation. Within the few past decades, wind turbine technology has become quite mature and is moving forward with a fast-growing rate. According to global wind energy report [1], installed wind energy capacity in the world doubles every three years. However, as the suitable places for onshore wind turbines are filled up, wind turbine installations shift offshore. Higher wind speeds and higher use rate also increase the attraction for offshore installations. In [2], it is predicted that a majority of wind turbines will be installed offshore in the next 20 years. However, lower availability due to unexpected failure is a major drawback for offshore wind turbines. It is more difficult to access offshore wind turbines compared to onshore turbines, so their maintenance and repair costs are higher. To overcome this problem, reliable concepts and components should be used in offshore wind turbine systems to reduce the need for regular maintenance and repair [3].

Direct Drive (DD) and geared drive are two main power take-off concepts for wind turbines. The direct drive system offers higher reliability as the mechanical gearbox, which requires regular lubrication and maintenance, is eliminated. Furthermore, mechanical losses in the gearbox is also eliminated, which increases the potential energy yield [4]. However, direct drive wind turbine system is challenged by the large and heavy generator, which introduces serious difficulties in both construction and installation stages, especially for multi-MW wind turbines [5]. One solution to cope with this issue is the use of high torque density generators to reduce the generator size; however, reliability should not be sacrificed to achieve higher torque density. Hence, both torque density and reliability

should be taken into account as critical factors when choosing a generator topology for offshore wind turbine applications.

To improve torque density, various topologies of electrical machines are proposed for DD wind turbine generators in the literature, which can be classified into two main categories, Permanent Magnet Synchronous Machines (PMSM) and Electrically Excited Synchronous Machines (EESM). There is great deal of flexibility in the geometry of PMSMs so that different configurations of PMSMs, including Radial Flux Permanent Magnet (RFPM), Axial Flux Permanent Magnet (AFPM), Transverse Flux Permanent Magnet (TFPM) and Permanent Magnet Vernier (PMV) or magnetically geared machines are discussed and evaluated for DD wind turbine applications [6–13]. The PMSMs promise higher efficiency and energy yield, higher torque density than the EESM owing to the absence of field winding. However, EESMs are more cost-effective than PMSMs due to the absence of rare-earth PMs. Economic, environmental and geopolitical issues like magnet price, depletion of magnet resources and concentration of rare-earth magnet resources in china are the main concerns regarding usage of rare-earth magnets in PMSM [14]. In EESMs, maximum allowable temperature is limited by insulation class, while in PMSMs both demagnetization of PM and insulation put restriction on operating temperature, hence, EESMs are potential of operating at higher temperatures. Furthermore, output voltage can be fully controlled once the wind turbine is equipped with an EESM. In addition, field flux can be controlled at different power to minimize generator loss and maximize annual energy yield [9]. PMSMs may be considered more reliable than EESM thanks to the absence of brushes and slip rings, while the risk of PMs demagnetization at high temperature and harsh atmospheric conditions decreases PM generators reliability.

Magnetically geared or Vernier machines have recently gained lots of attention because of their potential high torque density. Field excitation of Vernier machines can be either PM-based or electrically excited. Owing to magnetic gearing effect and flux modulation poles, mechanical speed is multiplied by the magnetic gear ratio so that a higher frequency MMF (magneto-motive force) is produced [15]. This phenomenon makes Vernier machines a suitable option for low-speed high-torque applications. Various topologies are proposed for permanent-magnet Vernier (PMV) machine [16–19]. In [16], a novel dual PM Vernier machine is proposed in which PMs are placed on both stator and rotor sides to increase fundamental value of the air gap flux density. In [17], a high-power factor dual stator single rotor Vernier machine is proposed. Although both the power factor and the torque density are improved in that topology with respect to single stator option, complex mechanical structure makes it less reliable. In [18], a magnetically geared generator called Pseudo Direct Drive (PDD) is proposed for direct drive wind turbine applications. This generator promises a high torque density and reasonable power factor; however, it is composed of two rotating parts, which make the structure complex and less robust to be used as an offshore wind turbine generator. Magnetically geared PMV machines require several times more PM compared to conventional DDPM generators, which makes them less attractive. For instance, the topology presented in [18] requires excessive mass of PM which is about 25% of the generator active materials mass. Using field winding instead of PMs may be considered as a solution to reduce the capital cost and make Vernier machines economically feasible for large scale wind turbines. For that purpose, the proposed generator in [18] is revised in [20] and PMs in high speed rotor are replaced with field windings to reduce PM usage.

In this paper, a novel Electrically Excited Claw Pole Vernier (EECPV) generator is proposed for Direct-Drive wind turbine applications. This topology adapts claw pole rotor with a single loop field winding and a conventional laminated stator carrying three phase winding. Due to the magnet-free structure, unlike Permanent Magnet Synchronous Machines (PMSM), it does not suffer from high and varying permanent magnet price [14]. Thanks to the rotor claw pole structure, the generator has a single loop field winding. As results of such a simple field winding, the proposed generator has lower field copper loss and higher efficiency compared to conventional EESM machines. Furthermore, a new structure is devised for the rotor which makes it possible to construct the rotor from laminated steel and get rid of the problems like, difficult manufacturing process and poor magnetic characteristic,

associated with the soft magnetic composite (SMC) materials [21,22]. To achieve higher torque densities, the number of rotor claw poles and stator teeth are designed so that magnetic gear effect is created and the generator benefits from frequency multiplying effect [23]. In addition to these advantages, the proposed generator has a reliable and robust structure for offshore wind turbines, because of its single rotating part. The paper is organized as follows. In Section 2, the proposed generator topology is introduced and its operating principle is discussed. An analytic-numeric design procedure is developed and explained in Section 3. In Section 4, a simplified parametric sweep is combined with the developed design procedure and design parameters maximizing the generator torque density is obtained. The geometrical and performance parameters of the final design are presented and the mechanical structure design is included. Finally, in Section 5, a conclusion is made based on the provided results. Enercon 7.5 MW, 12 rpm wind turbine generator (Enercon GmbH, Bremen, Germany) is chosen as the reference design and the proposed EECPV generator is designed for the same specifications.

## 2. Topology and Operation Principle

### 2.1. Generator Topology

The proposed EECPV generator is composed of a single stator and a single rotor carrying circular field winding. The generator topology is shown in Figure 1. The stator has a toothed structure with a three-phase distributed armature winding similar to the conventional EESMs. Open slots are used in the stator to modulate the air-gap flux and create magnetic gearing effect.

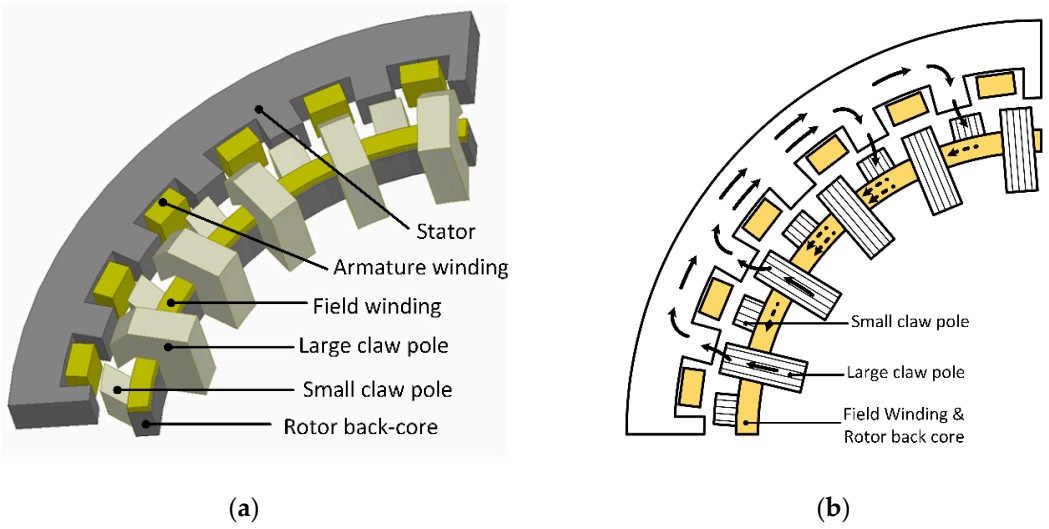

(**a**)           (**b**)

**Figure 1.** Proposed EECPV generator: (**a**) 3D representation under a pole pair; (**b**) 2D top view representation.

The rotor adopts claw pole structure with a single-loop field winding. The rotor structure is shown in Figure 2. As seen in Figure 2a, the rotor consists of four main parts, large claw pole, small claw pole, rotor back core and field winding. The field MMF magnetizes the claw poles in opposite directions and make them flux modulation poles of the rotor side. The rotor back core provides a flux path between opposing claw poles. A simplified 2D view of the flux path is schematically illustrated in Figure 2c. Thanks to the rotor claw pole structure, the field winding is composed of a single loop which is surrounded by the rotor parts. Reduced field copper loss compared to the conventional salient pole EESMs is one of the main advantages of the proposed topology. Furthermore, by adopting claw pole structure, rotor pole pitch does not impose any restriction on the available space for the field winding, while in conventional EESM pole pitch should be large enough to have sufficient space for field excitation windings.

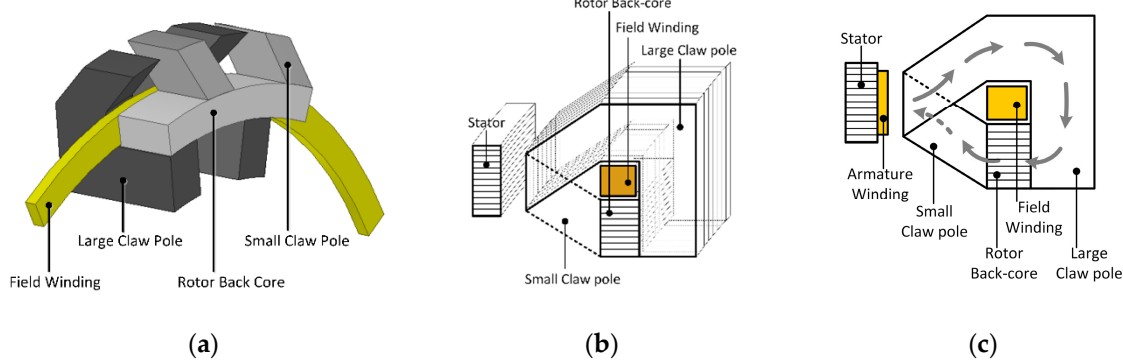

**Figure 2.** Rotor structure of the Proposed EECPV generator: (**a**) 3D representation of the rotor; (**b**) Representation of back core and claw poles laminations confront; (**c**) 2D cross-sectional view representation.

From the manufacturing point of view, it would seem to be more straightforward and easier if the rotor is designed similar to the proposed topology in [24], in which opposing claw-poles are identical and rotor back core is located behind them to create a pass for the flux. However, in that topology, the magnetic flux travels both in axial and tangential directions in the rotor back-core which makes it compulsory to use soft magnetic composite (SMC) material in the rotor back core [21]. SMC materials have poor magnetic characteristic compared to the laminated steels. Reduced mechanical strength is the major challenge with SMC materials. In addition, their manufacturing process is more difficult and expensive for large machines. However, when the rotor back core is placed in the position shown in Figure 2, where magnetic flux moves only in a 2D plain, it is possible to use laminated steel for the rotor back core instead of SMC material and get rid of the aforementioned drawbacks. In the proposed topology different stacking directions are used; that is the back core and the claw pole laminations confront each other in a perpendicular way. Therefore, the rotor stacking factor is reduced. Figure 2b shows how the rotor back core and claw pole laminations are positioned. In this configuration, the rotor stacking factor is obtained by multiplying stacking factors of the back core and claw poles. Lamination directions of different parts of the generator are shown with hatches in Figure 2b. The stator and the rotor back core are laminated in the axial direction, while the claw pole laminations are in the tangential direction.

*2.2. Operation Principle*

The operation of EECPV is similar to the conventional synchronous machines. The main difference is the presence of magnetic gearing effect which modulates the air gap flux density and creates frequency multiplying effect. A simplified representation of magnetic flux (leakage flux are not shown) under a pole is depicted in Figure 1c. Figure 3 shows the magnetic gearing effect in a 4-pole EECPV generator under a pole pair more clearly. As seen in Figure 3a, axis of the magnetic flux is vertical for the given rotor position. Once the rotor is rotated by half rotor pole pitch, axes of the magnetic flux are displaced by 90 electrical degree (45 mechanical degree) as shown in Figure 3b. In other words, for a small rotation of the rotor, there is a large flux displacement in the stator.

To create magnetic gear effect the following relationship should be satisfied [15],

$$Z_s = P_r + P_s \tag{1}$$

where $Z_s$ is the number of stator teeth, $P_r$ is the number of claw pole-pairs and $P_s$ is the number of armature winding pole-pairs.

The corresponding relationship between shaft speed and frequency of the magnetic field in the armature are expressed for conventional machines and Vernier machines in (2) and (3), respectively.

$$f_c = \frac{nP_s}{60} \tag{2}$$

$$f_v = \frac{nP_s}{60} \times G_r \tag{3}$$

where $n$ is the shaft speed, $G_r$ is called gear ratio which is expressed as follows,

$$G_r = \frac{P_r}{P_s} \tag{4}$$

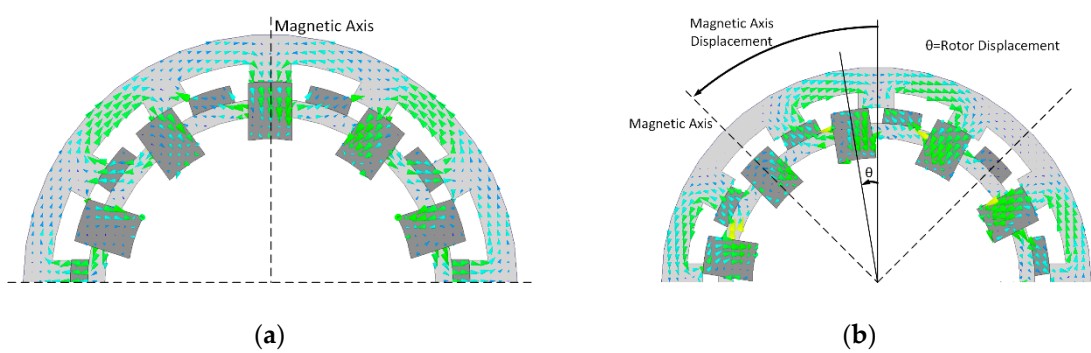

(**a**)  (**b**)

**Figure 3.** Representation of magnetic gear effect in a 4-pole EECPV generator: (**a**) Initial rotor position; (**b**) Displaced rotor position by $\theta$.

According to (2) and (3), the frequency of magnetic field in a Vernier machine is $G_r$ times higher than the magnetic field frequency of conventional machines, which helps to increase the torque density. Figure 4 depicts flux density distribution under a pole pair in the air gap of the proposed EECPV generator with a gearing ratio ($G_r$) of 11. Due to the magnetic gearing effect, the air gap flux density contains considerably large higher order harmonic. Consequently, magnetic loading in Vernier machines cannot be as large as conventional machines. Although this may be considered as the main drawback of Vernier machines, higher electrical frequency ensures the induced voltage magnitudes are higher compared to conventional machines.

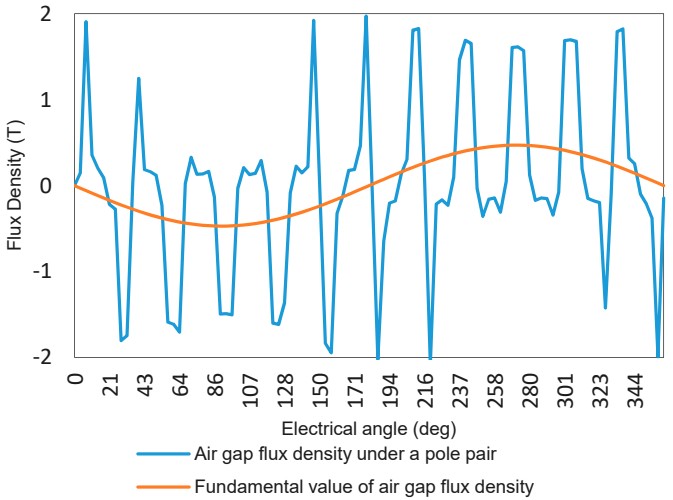

**Figure 4.** No load flux density distribution in the air gap of EECPV generator.

The higher order harmonic flux component in the air gap is mostly filtered by the stator teeth so that it does not travel through the stator back core and link the phase windings. Therefore, flux linkage and induced EMF are expected to be sinusoidal. Although the higher order harmonic flux does not link the phase winding, it travels in the rotor and stator teeth and causes extra core losses. Since, the generator is intended to be used for a low speed application (DD wind turbine), core losses are matter of concern and copper loss is the dominant loss component. It should be noted the effect of higher order harmonic is taken into account in core losses calculation by Finite Element Method (FEM) simulations.

## 3. Design Procedure of EECPV Generator

In this section, the developed design methodology for the proposed generator is discussed. The applied design methodology is an analytic-numeric method which is a combination of analytic equations and FE method. Since the main aim of this paper is to introduce topology of the proposed generator and show its potential sutibility for Direct-Drive wind turbine applications, the applied design procedure is kept as simple as possible. The detailed design procedure will be disscused in a separate paper. Torque expression of the EECPV generator can be expressed in terms of its main dimensions as,

$$T_e = \frac{\sqrt{2}}{8}\pi^2 k_w \times G_r \times q B_g D_g^2 L_{stk} \tag{5}$$

where $D_g$ and $L_{stk}$ are bore diameter and axial length of the generator, $B_g$ and $q$ are magnetic loading and electric loading and $k_w$ is the winding factor. It is well-known that choice of electric loading ($q$) depends on the cooling type and thermal behavior of the generator.

$B_g$ is defined as the peak pole flux divided by pole area. In conventional machines such as EESM, $B_g$ can be analytically calculated using equivalent magnetic circuit; however, the magnetic circuit of the EECPV machine is too complex and it is difficult to obtain an accurate analytical model for calculating $B_g$. Therefore, 3D Finite Element (FE) analysis is used to have an accurate estimation of $B_g$.

According to (5), $B_g$ is required for calculating generator main dimensions; however, the main dimensions and geometrical parameters of the generator are required for building an FE model and calculating $B_g$. Therefore, a set of design parameters is specified as a design vector which has adequate information to establish a primary FE model. Parameters of the design vector are introduced in Table 1.

**Table 1.** Parameters of the chosen design vector.

| Parameter | Definition |
|-----------|------------|
| $P_s$ | Number of pole pairs |
| $F_m$ | Field winding MMF |
| $a_{cp}$ | claw-pole arc ratio |
| $D_g$ | Bore diameter |
| g | Air gap length |
| $G_r$ | Gear ratio |
| $s_o$ | Stator slot opening ratio |
| $q$ | Armature Electric loading |
| $J_a$ | Current Density |

The generator geometrical parameters, such as tooth width, $t_w$, tooth height, $h_t$ and claw-pole width, $w_{cp}$ are calculated using (6), (7) and (8) using design vector parameters given in Table 1.

$$t_w = \frac{\pi D_g(1 - s_o)}{Z_s} \tag{6}$$

$$h_t = 0.5\left(\frac{Z_s t_w}{\pi} - D_g + \sqrt{\left(\frac{Z_s t_w}{\pi} - D_g\right)^2 + \frac{4 D_g q}{J_a k_{cu}}}\right) \tag{7}$$

$$w_{cp} = \frac{\pi(D_g - 2g)}{2P_r} \times a_{cp} \tag{8}$$

Using the given design vector and the calculated geometrical parameters, a primary FE model is established. Magnetostatic FE simulations are used to obtain $B_g$. Adaptive mesh which is controlled by the FEM software is used to for mesh generation. The stator back core height, $h_{sbc}$ and the rotor back core, $h_{rbc}$ are calculated using (9).

$$h_{sbc} = h_{rbc} = \frac{\pi D_g B_g}{4P_s B_{sat}} \tag{9}$$

where, $B_{sat}$ is the saturation flux density of the core material.

Number of turns per phase is calculated using (10),

$$N_{ph} = \frac{E_a}{4.44 f_v B_g A_p} \tag{10}$$

where, $E_a$ is induced EMF and $A_p$ is the area under a pole.

For a given design vector the design procedure is summarized in the following steps:

- Choose an initial value for axial length, $L_{stk}$.
- Calculate $h_t$, $t_w$, $w_{cp}$ (shown in Figure 5) analytically using (6), (7) and (8) in the terms of design vector and design constants (see Table 2).
- Establish the primary 3D FE model and obtain $B_g$
- Calculate values of $h_{sbc}$ and $h_{rbc}$ (shown in Figure 5) analytically using (9) in the terms of estimated $B_g$ and the design vector
- Calculate number of turns per phase using (10).
- Calculate $L_{stk}$ using (5) for a desired torque.
- Repeat this procedure until $L_{stk}$ is converged.

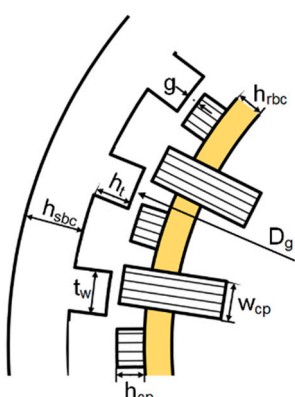

**Figure 5.** No load flux density distribution in the air gap of EECPV generator.

**Table 2.** Constant values assigned to the design vector.

| Parameter | Value |
|---|---|
| Bore diameter, $D_g$ | 7 m |
| Air gap length, $g$ | 5 mm |
| Gear ratio, $G_r$ | 11 |
| Stator slot opening ratio, $s_o$ | 0.6 |
| Armature Electric loading, $q$ | 70 kA/m |
| Current Density, $J_a$ | 2 A/mm$^2$ |
| Field current density, $J_f$ | 2 A/mm$^2$ |

## 4. EECPV Generator for Large Direct-Drive Wind Turbines

The EECPV generator is designed for the specifications same the Enercon 7.5 MW, 12 rpm generator as a comparison reference. Since total mass of the generators are compared, torque density is chosen as the desgin objective which should be maximize. The proposed design methodology in Section 3 is applied. To maximize the torque density, optimum design vector should be found through an optimization procedure. It is not within the scop of this paper to provide and discuss a detailed optimization procedure resulting in the global optimum. Therefore, a simplified parametric sweep is used to maximize the generator torque density.

### 4.1. Determination of the Optimum Design Vector

The specified design vector contains 10 parameters. If the parametric sweep is applied to all the variables in the design vector and corresponding torque density is calculated, a high computational effort will be required. It is worth noting that obtaining the most optimum EECPV generator for the desired application is not the main concern of this study. Therefore, a smplified parametric sweep is applied. To simplify the parametric sweep and reduce computational effort, only the parameters which are related to the rotor structure are swept. Thefore, only $P_s$, $F_m$, $a_{cp}$, are swept through specific ranges and the rest of the design vector parameters are assigned constant values reported in Table 2. The reported constant values in Table 2 are selected based on the author's experience. For each design vector, the EECPV generator is designed using the design procedure presented in Section 3. Based on the design procedure, FE simulation is used to calculate $B_g$ and rest of the calculations are performed analytically. Then, the design with maximum torque density is chosen as the optimum one. VacoFlux 50 (Vacuumschmelze GmbH, Hanau, Germany ) is used as the core material [25].

Figure 6 shows the obtained torque density with variation of field winding MMF ($F_m$) and claw-pole arc ratio ($a_{cp}$), when the number of pole pairs ($P_s$) is 6. As seen in the figure, maximum torque density is achieved once $F_m$ and $a_{cp}$ are 11 kA·t and 0.65, respectively.

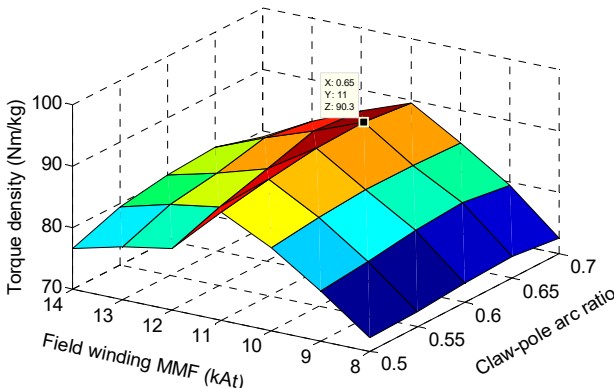

**Figure 6.** Variation of the EECPV generator torque density with field winding MMF ($F_m$) and claw-pole arc ratio ($a_{cp}$) for $P_s$ = 6 (Max. point: $F_m$ = 11 kA·t, $a_{cp}$ = 0.65, Torque density = 90.3 N·m/kg).

The maximum torque density, presented in Figure 6, is obtained for a constant number of pole pairs. However, it is necessary to sweep the number of pole pairs as well to obtain the optimum design vector yielding higher torque density. To reduce the number of required 3D FEM simulations, $F_m$ and $a_{cp}$ are assigned to be the obtained values in Figure 6, and only $P_s$ is swept. Again for each design vector, the generator is designed using the design procedure presented in Section 3. Then the corresponding torque density is calculated for each design vector. Figure 7 shows the calculated torque density versus $P_s$ variation. As it is seen in the figure, the optimum $P_s$ is 6. So the maximum torque density is achieved as 90.3 Nm/kg when $P_s$ = 6, $F_m$ = 11 kA·t and $a_{cp}$ = 0.65.

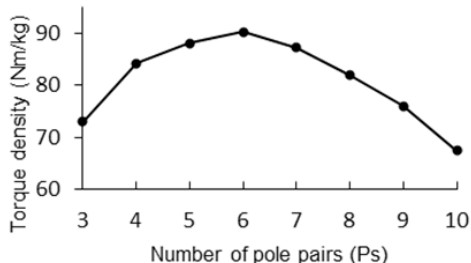

**Figure 7.** Variation of maximum torque density versus number of pole pairs.

### 4.2. Designed EECPV Generator

The EECPV generator is designed using the proposed design procedure in Section 3 and the obtained optimum design vector in Section 4.1. The design procedure is a combination of analytic equation and FE simulation. Geometrical parameters of the designed EECPV generator are presented in Table 3. The outline dimensions of the designed generator are depicted in Figure 8.

**Table 3.** Dimensions and parameters of the designed EECPV generator.

| Design Parameter | Value |
| --- | --- |
| Bore diameter, $D_g$ | 7 m |
| Stator axial length, $L_{stk}$ | 0.36 m |
| Outer diameter | 7.5 m |
| Pole pairs, $P_s$ | 6 |
| Claw-pole arc ratio, $a_{cp}$ | 0.65 |
| Tooth width, $t_w$ | 122 mm |
| Tooth length, $h_t$ | 95 mm |
| Outer stator back core length, $h_{sbc}$ | 116 mm |
| Claw pole length, $h_{cp}$ | 230 mm |
| Claw pole width, $w_{cp}$ | 108 mm |
| Rotor back core length, $h_{rbc}$ | 127 mm |
| Number of turns per phase, $N_{ph}$ | 192 |
| Filed winding turns, $N_f$ | 100 |
| Number of stator teeth | 72 |
| Number of claw pole-pairs | 66 |
| Copper mass | 18.8 t |
| Rotor mass | 33.9 t |
| Stator mass | 11.7 t |
| Total active mass | 64.4 t |

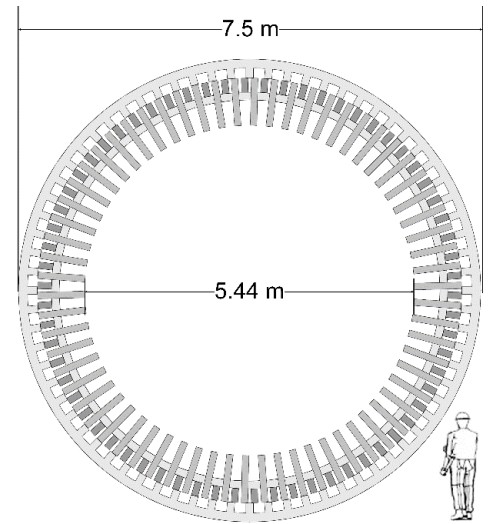

**Figure 8.** Outline dimensions of the designed EECPV.

To conform the design procedure, the designed generator is analyzed using Finite Element Method (FEM). 3D FE model of the designed generator is illustrated in Figure 9a. The no-load flux density distribution obtained from FE analysis is indicated in Figure 9b. Line to line induced EMF waveforms of the generator at rated speed are shown in Figure 10. As seen in the figure, the induced EMF magnitude is 4.6 kV which corresponds to RMS value of 3.3 kV as it is specified in Table 2.

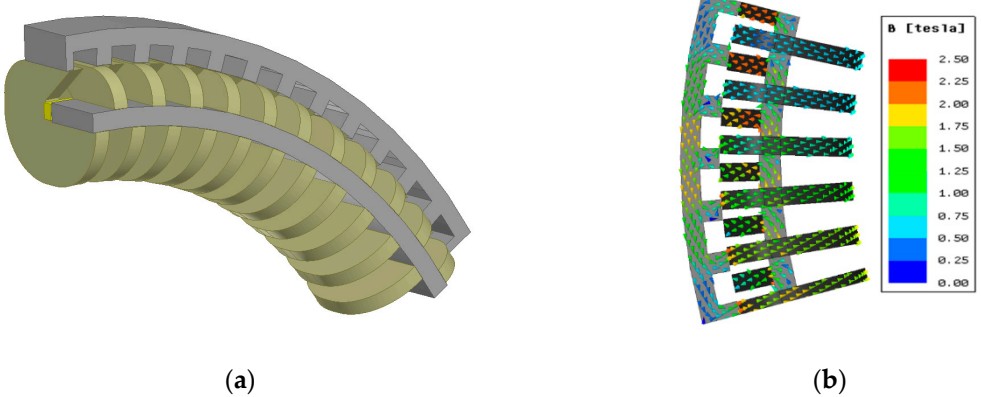

| (**a**) | (**b**) |

**Figure 9.** FEM model of the EECPV: (**a**) 3D FEM model representation; (**b**) No load flux density vectors.

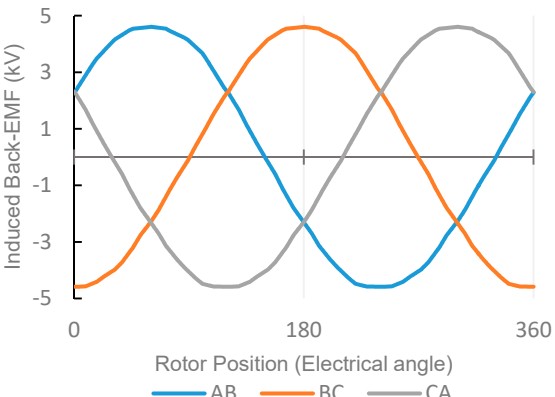

**Figure 10.** Line to line induced back-EMF waveforms of the EECPV generator in 12 rpm.

The rated electromagnetic torque of the designed generator obtained from FEM is indicated in Figure 11. The performance parameters of the designed generator for the rated operating condition obtained from FE analysis are presented in Table 4. As given in the table, the designed EECPV generator has considerably low power factor than the conventional generators. This is an expected result, due to the existence of the magnetic gear effect. Low power factor is an inherent drawback of the electrical machines with magnetic gear effect because of the lower fundamental flux density and higher leakage content in the air gap. Thanks to the specified low current densities and simple single loop field winding, efficiency of the designed generator is as high as 98% at the rated operation condition. However, it should be noted that only copper and core losses are taken into account in efficiency calculation and other loss components like mechanical losses and eddy currents loss in mechanical structure are neglected.

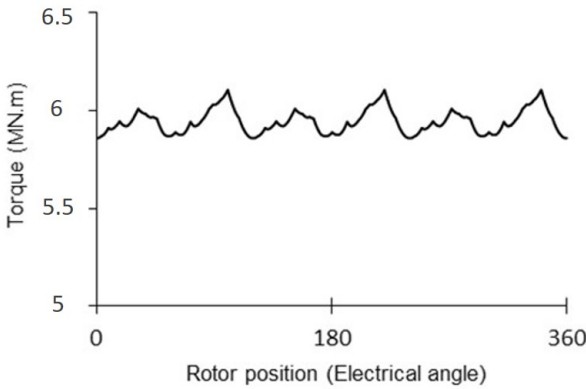

**Figure 11.** Electromagnetic torque waveform of the designed EECPV generator.

**Table 4.** Performance parameters of the designed generator obtained from FE analysis at rated load.

| Performance Parameter | Value |
|---|---|
| Phase current, rms | 1.32 kA |
| Line to line Induced EMF, rms | 3.3 kV |
| Torque | 6 MN·m |
| Field winding copper loss | 7.5 kW |
| Armature winding copper loss | 129.7 kW |
| Core losses | 12.5 kW |
| Power factor | 0.61 |
| Efficiency | 98.0% |

To acquire more insight about the designed generator, its performance is also evaluated for various operating conditions. The power curve and power coefficient of E-126/7.5 MW wind turbine manufactured by Enercon is illustrated in Figure 12. According to the manufacturer data, the rotational speed of the E-126 generator varies between 5 rpm and 12 rpm. However, the exact rotational speed is not specified for each operating point. Therefore, here, values between 5 rpm and 12 rpm, which are proportional to the generator power, are assigned as the generator rotational speeds. Figure 13 shows the assumed rotational speed versus the wind speed obtained from similar turbines in the literature.

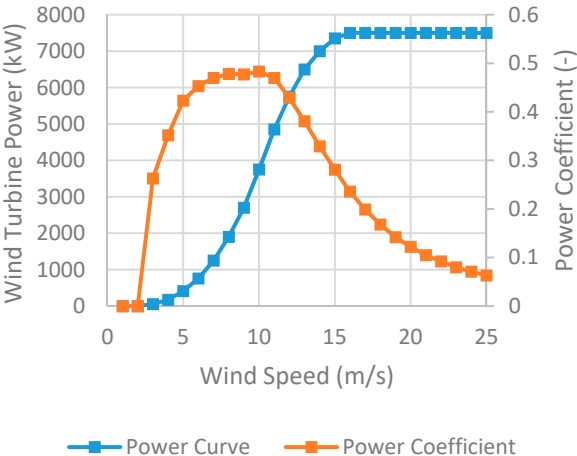

**Figure 12.** The power curve and the power coefficient of the Enercon E-126 wind turbine.

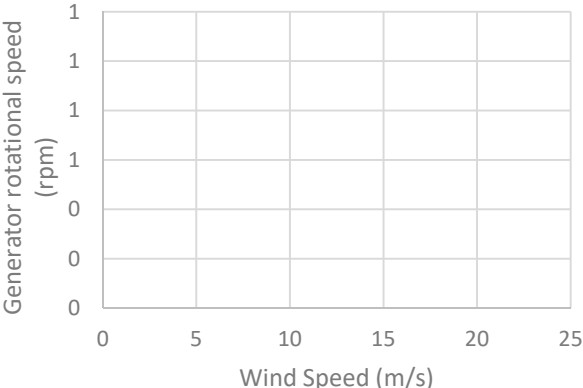

**Figure 13.** The assumed rotational speed versus wind speed for the generator of E-126 wind turbine.

Efficiency of the generator for the given operating range is shown in Figure 14. FE simulations are used to obtain the generator efficiency for each operating point. As seen in the figure, unlike the conventional Electrically Excited generators at which the efficiency at lighter load conditions is lower than the efficiency at rated load, the efficiency of the EECPV generator remains almost constant for a wide operating range, which helps to increase energy yield even at lower wind speeds. Such an interesting characteristic is because of single loop field excitation and reduced copper loss.

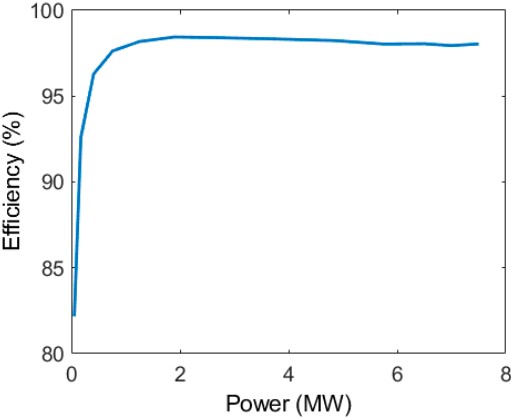

**Figure 14.** The efficiency of EECPV generator.

*4.3. Structural Mass Optimization*

Structural mass of a direct-drive generator is a major contributor to the overall mass, but it is usually ignored in electromagnetic designs and several papers just optimize active material mass. It is especially important in DDPMGs (Direct-Drive Permanent Magnet Generators) as the mechanical structure has to be stiff enough to keep the air gap clearance constant, which makes DD wind turbine generators mechanically demanding [26]. In [27], it is estimated that the structural mass of a 5 MW direct drive wind turbine generator is about 55% of the total mass.

In [28,29], a light-weight mechanical structure is proposed for the direct drive PM machines and an analytic optimization procedure is developed for minimizing the proposed structure weight. Figure 15 shows a simplified 2D view of the proposed mechanical structure. As seen in the figure, it is composed of hallow torque arms and a support frame. The hallow torque arms are responsible for transferring torque to the shaft, tolerating centripetal force and avoiding tangential deflection. The frame support should be designed in a manner that tolerates normal forces and does not allow normal deflection exceeds maximum allowable deflection which is 5% of the air gap length. According to the results provided in [28,29], the optimum number of torque arms is 5.

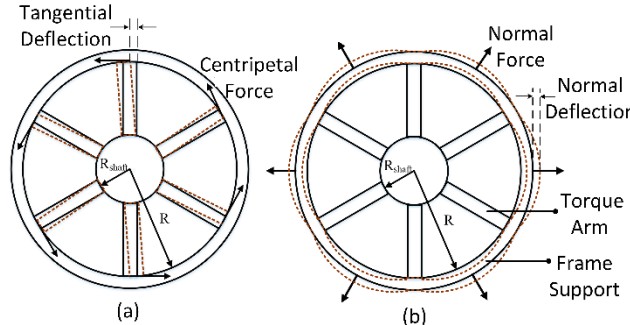

**Figure 15.** 2D representation of structural geometry: (**a**) Representation of centripetal force and tangential deflection; (**b**) Representation of normal force and normal deflection.

The proposed structure and the developed analytic calculation method in [28] are adapted for the EECPV generator to calculate the structural mass. The obtained results are presented in Table 5. Total structural mass of the proposed EECPV generator is estimated as 107.8 ton which is about 63% of the total mass.

**Table 5.** Structural and active material masses of the EECPV generator.

| Mass Component | Value | Percentage of Total Mass |
|---|---|---|
| Stator structural mass | 70.8 t | 41% |
| Rotor structural mass | 37.0 t | 21.5% |
| Copper mass | 18.8 t | 11% |
| Stator core mass | 11.7 t | 7% |
| Rotor core mass | 33.9 t | 19.5% |
| Total mass | 172.2 t | 100% |

*4.4. Comparison*

The mass and the torque density of the proposed generator are compared with a few commercial and conceptual direct-drive wind turbine generators in Table 6 [28–30]. Enercon E-126 is one of the largest (7.5 MW) commercially available direct-drive wind turbine, which uses a 220 t electrically excited synchronous generator. The Switch, a permanent-magnet machine manufacturer, is included in the comparison with their 3.8 MW, 21 rpm DDPM generator. NTNU (Norwegian University of Science and Technology) presented a conceptual DDPM generator design for a large (10 MW, 13 rpm) turbine. Similarly, two DDPM generator designs are included in the table by NREL (National Renewable Energy Laboratory) and AMSC (American Superconductor Corporation). From the table it can be seen that, the torque density of DDPM generators are around 25 kNm/t.

**Table 6.** Mass and torque density comparison of the proposed generator with the commercial and conceptual DD wind turbine generators.

| Generators | Type | Power (MW) | Speed (rpm) | Torque (MN·m) | Mass (t) | T/Mass (kN·m/t) |
|---|---|---|---|---|---|---|
| Proposed EECPV | EESG * | 7.5 | 12 | 5.97 | 172.2 | 34.7 |
| Enercon E-126 | EESG | 7.5 | 12 | 5.97 | 220 | 27.2 |
| The Switch [30] | PMSG | 3.8 | 21 | 1.73 | 81 | 21.6 |
| NTNU [31] | PMSG | 10 | 13 | 7.35 | 260 | 28.2 |
| NREL-AMSC [32] | PMSG | 6 | 12.3 | 4.66 | 177 | 26.3 |
| NREL-AMSC [32] | PMSG | 10 | 11.5 | 8.3 | 315 | 26.4 |

* Electrically Excited Synchronous Generator.

Furthermore, permanent magnet generators use high amount of magnet, which increases the overall cost due to volatile prices of rare-earth materials. Enercon E-126 does not use any permanent magnet as it uses an electrically excited field winding, similar to the proposed generator. Since the proposed EECPV generator is designed for the same specifications as Enercon E-126, it is fair to compare them in the terms of mass. The mass of the proposed generator is reduced by 48 tones with respect to Enercon E-126 (22% reduction). Thanks to magnetic gearing effect, the fundamental flux density in the air gap rotates $G_r$ times faster than the fundamental frequency. Although magnitude of the fundamental flux density in the EECPV generator is not as high as conventional electrical machines, frequency multiplying effect of existing magnetic gear compensates for the low fundamental flux density and results in a higher torque density.

**5. Conclusions**

A novel Electrically Excited Claw Pole Vernier (EECPV) generator is proposed for large offshore wind turbines in this paper. The rotor has a claw pole structure and a simplified ring-type field winding. Moreover, the machine structure does not require SMC materials as most claw-pole machines do. Owing to Vernier effect torque density is considerably improved up to 34.7 Nm/kg. Furthermore, absence of permanent-magnet simplifies the manufacturing process and reduces the overall cost of the generator. A FE-based design process is developed to verify the designs and the optimum design vector is selected by investigating the effect of design parameters on generator torque density. The optimized generator for a 7.5 MW, 12 rpm DD wind turbine application is 22% lighter than Enercon E-126 and is promising toward achieving larger offshore wind turbines.

**Author Contributions:** Conceptualization O.K. and R.Z.; Methodology O.K.; Validation & Analysis, R.Z.; Writing—Original Draft Preparation, O.K. and R.Z.

**Funding:** This research received no external funding.

**Conflicts of Interest:** The authors declare no conflict of interest.

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
