# Peer review of "A Rare-Earth Free Magnetically Geared Generator for Direct-Drive Wind Turbines"

_energies, doi:10.3390/en12030447_

Round 1
Reviewer 1 Report
This paper focus on the design of a 7.5 MW Vernier type magnetically geared direct-drive generator for large wind turbines which does not require the use of rare-earth permanent magnets.
The Reviewer proposes some modifications in order to improve the quality of the work.
1.- Introduction section. The problematic of rare-earth permanents must be introduced, which is clearly stated in
https://www.sciencedirect.com/science/article/pii/S136403211501504X
2.- Introduction section. The main novelties of this research work along with the main contributions should be stressed and compared with other similar state-of-the-art works.
3.- Table 1. The current density of 2 A/mm2 seems too low, the diameter of 7 m too large and the axial length very short, thus resulting quite odd aspect ratio. Why such odd design parameters?
4.- Table 6. EESG must be defined.
5.- The authors do not provide details about the process carried out to calculate the design parameters. It has been done by means of FEM? Without such details the paper is not worth to be published.
6.- The authors must detail the procedures applied to determine the results presented in Figures 6 to 14 and in Tables 2 to 6.
7.- Please include details about the possible differences between experimental results and calculations carried out in the paper, while trying to quantify such differences.
8.- This paper does not include any experimental results, which probably are the most suitable means to validate the developments and proposals done along the paper. It would be nice to see some experimental results, even with a smaller machine in order to prove that the results presented are accurate enough.
The Reviewer suggests revising the work based on the suggestions above in order to improve its quality.
Author Response
Thank you for your valuable comments. Here are our responses to your comments:
1.- Introduction section. The problematic of rare-earth permanents must be introduced, which is clearly stated in
https://www.sciencedirect.com/science/article/pii/S136403211501504X
Based on your comments, the PMs are discussed more in detail. Please refer to lines 54-56 for the modified version.
2.-
Introduction section. The main novelties of this research work along
with the main contributions should be stressed and compared with other
similar state-of-the-art works.
Introduction section is modified to emphasize the main contribution. Please refer to the lines 84-93 for the modified version.
3.-
Table 1. The current density of 2 A/mm2 seems too low, the diameter of 7
m too large and the axial length very short, thus resulting quite odd
aspect ratio. Why such odd design parameters?
Although, the current density is low, a high number of turns increases the electrical loading (and hence the copper loss), which is the case for large-diameter direct-drive wind turbine generators. The choice of such a large diameter (D g =7 m) is because of the fact that direct drive wind turbine generator should be designed in large diameters due to the high torque requirement. The axial length of the final design is 0.36 m, it should be noted that this is just the active length of the generator. The total axial length will be larger once mechanical structure is included. Such high aspect ratios are typical in MW range direct drive wind turbine generators.
4.- Table 6. EESG must be defined.
Corrected ( in line 340)
5.- The authors do not provide details about the process carried out to calculate the design parameters. It has been done by means of FEM? Without such details the paper is not worth to be published.
Thank your for your comments. More details are added to the Sections 3, 4 and 4.1 to clarify how the design parameters are chosen and design methods.
6.- The authors must detail the procedures applied to determine the results presented in Figures 6 to 14 and in Tables 2 to 6.
More details are included in Section 4.2 to clarify the provided results.
7.- Please include details about the possible differences between experimental results and calculations carried out in the paper, while trying to quantify such differences.
8.- This paper does not include any experimental results, which probably are the most suitable means to validate the developments and proposals done along the paper. It would be nice to see some experimental results, even with a smaller machine in order to prove that the results presented are accurate enough.
7-8:
Thank you very much for your precious comments. We also believe experimental results would improve the quality of the paper, however we had faced two challenges: the first one is the lack funding for manufacturing a prototype, and the second one is even if we could have the funding, it probably would be in the kW power range. It would prove the concept, however due to scaling issues between a kW range machine and MW range full-scale machine (such as non-linear increase in the structural mass due to increased diameter or necessity to increase the air-gap clearance ratio due to manufacturing tolerances at a larger diameter, and reduction in power factor), it probably wouldn’t give enough information on power density or total mass of a MW-scale prototype. Therefore, initially we aimed to evaluate the feasibility of the proposed novel topology for the wind turbine industry by comparing it with commercial alternatives.
In the revised version, we expanded our simulations to include thermal and power factor effects and also the section on structural mass is expanded and a detailed comparison with existing commercial generators is presented. We hope that would be enough for a fair comparison and presentation of the proposed novel topology at this stage.
Reviewer 2 Report
1. The description of the state of the art is poor. More and recent
publications in the field should be taken into account.
2. The proposed structure is interesting.
3. If you planned to have a section about the design, you should provide
all the details, not only two sizing equations. This way the section is
without any use.
4. The results in Figs. 6 and 7 seems to be obtained by means of
simulations, but not a word can be found about the program or
methodology used for it. Had you been using the same FEM based program
mentioned later?
5. You are using the term "optimized generator", without mentioning
anything about the optimization process: what was the objective
function, the optimization method used, etc. These details mandatory has
to be added to the paper.
6. I much appreciate that structural mass computations were included in
the paper. These indeed are important. Unfortunately, the paper lack
details concerning these computations, Moreover, my above observations
related to the optimization are valid also here.
Author Response
Thank you for your valuable comments. Here are our responses to your comments:
1. The description of the state of the art is poor. More and recent
publications in the field should be taken into account.
The introduction section has been modified with inclusion of the recent publications in the literature. All changes can be seen in the attached file.
2. The proposed structure is interesting.
Thank you for your comments.
3. If you planned to have a section about the design, you should provide
all the details, not only two sizing equations. This way the section is
without any use.
Since the main objective in this paper is to introduce topology of the proposed generator and show its potential suitability for DD wind turbine application, the focus of the paper is focused on the general topology. However, detailed specifications of the proposed machine is presented in the following sections.
4. The results in Figs. 6 and 7 seems to be obtained by means of
simulations, but not a word can be found about the program or
methodology used for it. Had you been using the same FEM based program
mentioned later?
The provided results in Fig. 6 and Fig. 7 are obtained based on the design process presented in Section 3. Details of the simulations are presented now in Section 3.
5. You are using the term "optimized generator", without mentioning
anything about the optimization process: what was the objective
function, the optimization method used, etc. These details mandatory has
to be added to the paper.
In this paper, parametric sweep is used for torque density maximization. Since optimization procedure is not the main focus of this paper at this stage, a simplified parametric sweep has been applied, as a result only a local maximum of torque density is achieved. Based on your comments, the ‘optimized generator’ term in the paper is replaced with the ‘designed generator’ term and the paper is modified to clarify the methods and design parameters used.
6. I much appreciate that structural mass computations were included in
the paper. These indeed are important. Unfortunately, the paper lack
details concerning these computations, Moreover, my above observations
related to the optimization are valid also here.
Thank you for your comments. Indeed, we believe that structural mass is important for a fair comparison between different topologies. However, the focus of this paper is not to develop a mechanical mass optimization tool, therefore the model developed by Zavvos et al. for direct-drive wind turbines has been utilized in this paper, and details of the used mechanical model can be found in the cited documents.
Reviewer 3 Report
Dear Authors,
I have some comments to your article:
1. Lack at the end of the Introduction how an article is organized.
2. All units, indexes in symbols in text and equations should be checked carefully.
3. Line 163 Finite Element Analysis (FEA) simulations and line 225 – FE – rather FEM.
4. It would be good to give next steps how the equation (5) and (6) were obtained.
5. Line 181 do 189 - it would be good to illustrate the design procedure in the block diagram of the algorithm.
6. As seen in the figure, it is composed of five hallow torque arms and a support frame. There are six in the Figure 15.?
7. Please consider whether with six arms you can’t get bumps and quite big vibrations? Despite the low speed there may be significant amplitudes of vibrations of specific frequencies resulting from the relation of the number of poles and arms.
8. The number of cited literature should be increased, especially in literature from the last 18 months.
Author Response
Thank you for your valuable comments. Here are our responses to your comments:
1. Lack at the end of the Introduction how an article is organized.
A section summarizes the paper has been included in the introduction section.
2. All units, indexes in symbols in text and equations should be checked carefully.
Typos in the units and equations are corrected.
3. Line 163 Finite Element Analysis (FEA) simulations and line 225 – FE – rather FEM.
Corrected.
4. It would be good to give next steps how the equation (5) and (6) were obtained.
We could include these steps, but we are limited by the number of pages, we had to shorten the paper. The equation is not specific to the proposed topology but just a variation of the well-known equation used for the general sizing of electrical machines.
5. Line 181 do 189 - it would be good to illustrate the design procedure in the block diagram of the algorithm.
Thank for your comment. We initially tried to represent it as block diagram, however, as the procedure is quite straightforward (without any decision branches, or iterations), we decided to represent it in bullet-point to better utilize the text-space.
6. As seen in the figure, it is composed of five hallow torque arms and a support frame. There are six in the Figure 15.?
Figure 15 is just used to illustrate the torque components and structure components. A five-arm structure is used for mechanical mass calculations.
7. Please
consider whether with six arms you can’t get bumps and quite big
vibrations? Despite the low speed there may be significant amplitudes of
vibrations of specific frequencies resulting from the relation of the
number of poles and arms.
Yes, you are right. According to the results provided in [22] and [23], the optimum number of torque arms (both for total mass and for minimization of torque harmonics) is selected as five. The designed mechanical structure in the paper has also five torque arms.
8. The number of cited literature should be increased, especially in literature from the last 18 months.
Based on your comments, several recent publications are added to the paper as can be seen from the attached manuscript.
Reviewer 4 Report
This paper proposes a rare-earth free magnetically geared generator. The proposed generator is compared with some generators. My comments are as follows.
1. The difference of T/Mass in Table 6 between the proposed EECPV and Enercon E-126 is large. The reason must be discussed. Are the same current density and air gap length with Enercon E-126 used in the proposed model?
2. The number of stator teeth and rotor poles of the designed EECPV shown in Figure 8 should be shown in Table 3.
3. The power factor of the designed EECPV is low. The reason should be discussed.
Author Response
Thank you for your valuable comments. Here are our responses to your comments:
1. The difference of T/Mass in Table 6 between the proposed EECPV and
Enercon E-126 is large. The reason must be discussed. Are the same
current density and air gap length with Enercon E-126 used in the
proposed model?
Based on your comments, a discussion is added to the paper. Refer to the lines 346-350. Unfortunately, there is no information available in literature about the current density and air gap length of the Enercon E-126. The typical current density value is 3 A/mm2 for air-cooled electrical machines. In this paper, a conservative current density value of 2 A/mm2 is chosen for both the stator winding and the field winding.
2. The number of stator teeth and rotor poles of the designed EECPV shown in Figure 8 should be shown in Table 3.
Corrected.
3. The power factor of the designed EECPV is low. The reason should be discussed.
A discussion regarding the power factor of the proposed generator is added. Please refer to the lines 269 to 273.
Reviewer 5 Report
This paper presents the study on the magnetically geared generator without the use of permanent magnets.
To further improve the work, the authors may consider the following issues:
(1) The paper introduced the use of SMC material for claw pole type; this part may need more description, e.g. the design and analysis. Some good relevant references may be: Liu et al., “Design Issues for Claw Pole Machines with SMC Cores”, Energies, Vol. 11, No. 8, Article 1998, August 2018. Guo, et al., “Accurate Determination of Parameters of a Claw Pole Motor with SMC Stator Core by Finite Element Magnetic Field Analysis”, IEE Proceedings – Electric Power Application, Vol. 153, No. 4, pp. 568-574, July 2006.
(2) Some solid validation, e.g. experimental results on a prototype, if available, would be very useful.
Author Response
Thank you for your valuable comments. Here are our responses to your comments:
(1) The paper introduced the use of SMC material for claw pole type; this part may need more description, e.g. the design and analysis. Some good relevant references may be: Liu et al., “Design Issues for Claw Pole Machines with SMC Cores”, Energies, Vol. 11, No. 8, Article 1998, August 2018. Guo, et al., “Accurate Determination of Parameters of a Claw Pole Motor with SMC Stator Core by Finite Element Magnetic Field Analysis”, IEE Proceedings – Electric Power Application, Vol. 153, No. 4, pp. 568-574, July 2006.
In this paper, a new topology is proposed so that the generator can be manufactured using laminated steels, so use of 3D magnetic materials like SMC is not required. However, based on your comments, the paper has been added to the paper.
(2) Some solid validation, e.g. experimental results on a prototype, if available, would be very useful.
We also believe experimental results would improve the quality of the paper, however we had faced two challenges: the first one is the lack funding for manufacturing a prototype, and the second one is even if we could have the funding, it probably would be in the kW power range. It would prove the concept, however due to scaling issues between a kW range machine and MW range full-scale machine (such as non-linear increase in the structural mass due to increased diameter or necessity to increase the air-gap clearance ratio due to manufacturing tolerances at a larger diameter, and reduction in power factor), it probably wouldn’t give enough information on power density or total mass of a MW-scale prototype. Therefore, initially we aimed to evaluate the feasibility of the proposed novel topology for the wind turbine industry by comparing it with commercial alternatives.
Round 2
Reviewer 1 Report
The Authors have replied almost all my questions, although some improvements are still required.
1. Introduction section. The main novelties of this research work along with the main contributions should be further stressed and compared with other similar state-of-the-art works.
2. Section 3. The equations and FEM model require a detailed analysis, so that the reader can replicate the procedure. A flowchart can help as well as the details of all analytical equations applied and the FEM details (include mesh type, nodes, elements, type of mesh used in the air gap, a sketch with the mesh, etc.).
Author Response
Thank you for your valuable comments.
1. Introduction
section. The main novelties of this research work along with the main
contributions should be further stressed and compared with other similar
state-of-the-art works.
The last paragraph of the introduction section is thoroughly revised. The contribution of the paper and the novelty of the topology is emphasized and new papers from the literature are cited.
2. Section 3. The equations and FEM model require a detailed analysis, so that the reader can replicate the procedure. A flowchart can help as well as the details of all analytical equations applied and the FEM details (include mesh type, nodes, elements, type of mesh used in the airgap, a sketch with the mesh, etc.).
Section 3 is modified to include important design equqations and methodology is included. Additional information regarding the finite element model is also included in the section 3.
Reviewer 2 Report
Unfortunately I am not able to accept the most significant part of the responses of the authors, those regarding the description of the simulation and optimization program, and the structural computations. The authors states that these issues are not in the focus of the paper. This is hardly to be accepted, since the paper lacks also of experimental results. Upon my opinion a paper focusing only on a generator topology (even if that is novel and interesting) cannot be published in Energies.
Author Response
Thank you for your comments. We understand your concerns regarding the lack of experimental results. Although, we are not able to present a prototype at this stage, we improved the paper in the following ways:
The introduction section is throughly revised. The contribution of the paper and the novelty of the topology is emphasized and new papers from the literature are cited.
Section 3 is modified to include important design equations and methodology is included. Additional information regarding the finite element model is also included in the section 3.
Reviewer 4 Report
This paper is well revised.
Author Response
Thank your for your valuable comments and support.